# Metabolism and Pharmacokinetics of SP-8356, a Novel (1*S*)-(−)-Verbenone Derivative, in Rats and Dogs and Its Implications in Humans

**DOI:** 10.3390/molecules25081775

**Published:** 2020-04-13

**Authors:** Yuanyuan Zhou, Mun Hwan Oh, Yeon Joon Kim, Eun-yeong Kim, Jinhong Kang, Sung Chung, Chung Ju, Won-Ki Kim, Kiho Lee

**Affiliations:** 1College of pharmacy, Korea University, Sejong 30019, Korea; 2Research Headquarters, Shin Poong Pharm. Co., Ltd., Ansan, Gyeonggi 15610, Korea; 3Departments of Biomedical Sciences and Neuroscience, College of Medicine, Korea University, Seoul 02841, Korea; 4Institute of Inflammation Control, Korea University, Seoul 02841, Korea; 5Institute of Pharmaceutical Science and Translational Research, Korea University, Sejong 30019, Korea; 6Biomedical Research Center, Korea University Guro Hospital, Seoul 08308, Korea

**Keywords:** SP-8356, pharmacokinetics, metabolism, bioactivation, conjugation

## Abstract

(1*S*,5*R*)-4-((*E*)-3,4-dihydroxy-5-methoxystryryl)-6,6-dimethylbicylco[3.1.1]hept-3-en-2-one (SP-8356) is a novel (1*S*)-(−)-verbenone derivative that is currently in preclinical development for the treatment of ischemic stroke and atherosclerosis. This report aimed at characterization of the metabolism and pharmacokinetic properties of SP-8356. Following intravenous dose in rats and dogs, plasma concentrations of SP-8356 declined rapidly with high clearance (CL) and short half-life; after oral administration in both species, its plasma levels were below the quantitation limit. Fourteen circulating metabolites, formed by mono-oxygenation, demethylation, glucuronidation, catechol *O*-methylation, sulfation and oxidation (bioactivation) followed by glutathione (GSH) conjugation, were tentatively identified in both species. Urinary excretion of SP-8356 appeared to be minimal in rats, compared to its metabolites. GSH conjugate of SP-8356 was also formed during incubation with rat liver S9 fraction consistent with oxidative bioactivation; this bioactivation was almost completely inhibited by the cofactors for glucuronidation, sulfation and methylation, indicating that it may be abolished by competing metabolic reactions in the body. The human pharmacokinetics of SP-8356 was predicted to be similar to that of the animals based on the current in vitro metabolic stability results. In summary, rapid phase II metabolism appears to be mainly responsible for its suboptimal pharmacokinetics, such as high CL and low oral absorption. Because of competing metabolic reactions, potential safety risks related to SP-8356 bioactivation may be low.

## 1. Introduction

Verbenone, a bicyclic monoterpene ketone, is one of the main ingredients and the flavor source of a variety of plant essential oils, herbal teas and spices, such as *Rosmarinus officinalis*, *Verbena triphylla* L., *Chrysanthemum morifolium Ramat.* and *Piper aleyreanum* [1,2,3,4]. It is also released by bark beetles from the host tree-derived precursor α-pinene and plays the role of an anti-aggregation pheromone [5]. Verbenone-containing essential oils have been reported to have various biological activities that include antibacterial activity against oral pathogens, antinociceptive activity in formalin-induced licking actions in mice, anti-inflammatory activity in carrageenan-induced pleurisy in mice and antiulcer effect in acute ethanol-induced gastric lesions in rats [1,3]. One of the enantiomers (1*S*)-(−)-verbenone showed excellent acaricidal activity against house dust mites and elicited an anticonvulsive effect in pentylenentetrazol-induced seizures in mice by regulating RNA expression of the neurotrophin brain-derived neurotrophic factor, the transient proto-oncogene protein c-fos and cyclooxygenase 2 [6,7].

Over the last decade, many efforts have been made to synthesize (1*S*)-(−)-verbenone derivatives with improved efficacy and selectivity. A series of novel (1*S*)-(−)-verbenone derivatives that include (1*S*,5*R*)-4-((*E*)-3,4-dihydroxy-5-methoxystryryl)-6,6-dimethylbicylco[3.1.1]hept-3-en-2-one (SP-8356) have been reported to show anti-ischemic activity in oxygen–glucose deprivation/reoxygenation (OGD/R)-induced neuronal injury in rat cortical neuronal/glial co-culture model, by increasing heme oxygenase-1 expression in astrocytes [8]. Also, (1*S*,5*R*)-6,6-dimethyl-4-((*E*)-4-methylstyryl) bicyclo[3.1.1]hept-3-en-2-one (LMT-335) not only reduced OGD/R-induced injury by decreasing intracellular oxidative/nitrosative stress, it also potently inhibited N-methyl-D-aspartic acid-evoked excitotoxicity in rat cortical neurons [9]. Recently, it has been reported that SP-8356 exerted anti-breast cancer activities both in vitro (cell cycle arrestment and cancer cell migration inhibition) and in vivo (inhibition of tumor growth and metastasis in a mouse xenograft model of triple-negative breast cancer), through regulation of nuclear factor kappa B signaling and metastasis-associated gene expression [10].

Low plasma exposure of SP-8356 accompanied by significantly higher plasma level of its glucuronide metabolite was observed following intraperitoneal administration in mice, despite its apparent in vivo efficacy [10]. Therefore, it is necessary to characterize the disposition properties of the compound to better understand the relationships between its pharmacological activity and pharmacokinetics. It is also necessary to characterize the metabolism and pharmacokinetics in both rodents and non-rodents, in support of its development as a drug candidate.

In this report, the pharmacokinetic properties of SP-8356 were investigated following intravenous (*i.v.*) and oral (*p.o.*) administration in rats and dogs. Circulating and urinary metabolites were tentatively identified using a high-resolution LC-MS/MS system in the animals and the corresponding metabolic pathways were proposed. The metabolic stability of SP-8356 was examined using liver S9 fractions of the animals and humans, well-established in vitro system being widely used for drug metabolism studies [11], in the presence of relevant cofactors such as β-nicotinamide adenine dinucleotide 2′-phosphate (NADPH), uridine 5′-diphosphoglucuronic acid (UDPGA), 3’-phosphoadenosyl-5’-phosphosulfate (PAPS), *S*-(5′-adenosyl)-l-methionine (SAM) and glutathione (GSH). The possibility of bioactivation and the role of competing metabolic pathways in negating the bioactivation were explored as well. These results provide a mechanistic understanding of the pharmacokinetic characteristics of SP-8356 in these preclinical animal species and furnish useful information to anticipate its disposition and performance in humans.

## 2. Results

### 2.1. Pharmacokinetics of SP-8356 in Rats and Dogs

The pharmacokinetics of SP-8356 was investigated in male Sprague-Dawley (SD) rats and Beagle dogs following a single *i.v.* and *p.o.* administration.

Following the *i.v.* dose, plasma concentration of SP-8356 declined rapidly in both rats and dogs (Figure 1) with a half-life (*t*_1/2_) < 0.1 h (Table 1). Its systemic clearance (CL) was (21.0 ± 6.5 and 4.0 ± 0.3) L/h/kg in rats and dogs, respectively (Table 1), much higher than the hepatic blood flow of the animals [12]. The mean volumes of distribution during terminal phase (V_z_) and at steady state (V_ss_) ranged (1.3–1.9) L/kg (Table 1), far exceeding the total body water of the animals [12], suggesting that SP-8356 distributed well outside the vasculature in the body. Following *p.o.* dose in both rats and dogs, the plasma levels of SP-8356 were below its quantitation limit, that is, <10.5 ng/mL, throughout the 24 h time courses (Figure 1, Table 1).

### 2.2. In Vivo Metabolite Identification in Rats and Dogs

The metabolites of SP-8356 present in rat and dog plasma and rat urine samples obtained from the *i.v.* pharmacokinetics studies described in Section 2.1. were identified and characterized in this study using Q-TOF LC-MS/MS system.

A total of 14 putative metabolites were identified in rat and dog plasma. Table 2 summarizes their mass spectral data. The metabolite peaks designated M1 and M2 both had *m*/*z* value 16 u higher than that of the parent, indicating the addition of one oxygen (i.e., monooxygenation). M3 had *m*/*z* value 14 u lower than that of the parent, consistent with a metabolite formed by the loss of CH_2_ (i.e., demethylation). M4 and M5 exhibited mass shifts of (176 and 352) u, respectively, higher than that of the parent, suggesting that they were formed by mono- and di-glucuronation, respectively; both of them showed product ions of *m*/*z* (175, 113 and 85), characteristic of glucuronide moiety. M6 had *m*/*z* value 14 u higher than that of the parent, consistent with mono-methylation. M8 was assigned as a sulfate metabolite, as its *m*/*z* value was 80 u (SO_3_) higher than that of the parent. The *m*/*z* values of M7, M9 and M10 were consistent with methylation + glucuronidation (+190 u), methylation + sulfation (+94 u) and oxygenation+methylation (+30 u), respectively. The exact positions of the biotransformation that occurred to the parent molecule could not be assigned based on the mass spectral data available from this study. M11 had *m*/*z* value 305 u higher than that of the parent, indicative of GSH conjugation (+307 u) combined with dehydrogenation (−2 u); its product ions were also consistent with GSH conjugation, that is, *m*/*z* 331 (parent +32 u) and *m*/*z* 272 (GSH-35 u) for fragmentation of C–S bond between the parent and GSH and *m*/*z* 316 (product ion *m*/*z* 331–15 u) for demethylation of the product ion *m*/*z* 331. M12 and M13 with *m*/*z* values 475.1544 (+176 u) and 418.1330 (+119 u) were assigned as cysteinylglycine (Cys–Gly) and cysteine (Cys) conjugate, respectively; they both had product ions of *m*/*z* (331 and 316), consistent with the presence of a Cys residue. M14 had *m*/*z* value 42 u (+C_2_H_2_O) higher than that of M13, suggesting an *N*-acetylcysteine (AcNCys) conjugate.

Scheme 1 proposes the biotransformation pathway of SP-8356 in rats and dogs based on the current results. SP-8356 appears to be metabolized by mono-oxygenation (M1 and M2), demethylation (M3), glucuronidation (M4 and M5), catechol O-methylation (M6), sulfation (M8) and oxidation (bioactivation), followed by GSH conjugation (M11). M12–M14 can be formed by hydrolysis of the GSH moiety of M11. All other metabolites (M5, M7, M9 and M10) can be produced by combinations of the metabolic reactions mentioned earlier.

Figure 2 shows the plasma exposure-time profiles of the detected metabolites following a single *i.v.* dose of SP-8356. In both rats and dogs, the metabolites appeared rapidly in plasma immediately after the dose with *t*_max_ ≤ 1 h (Figure 2, Table 3). In rat plasma, the metabolites were detected only up to 1 h after the dose; whereas in dog plasma, they were present throughout the time course (Figure 3). In rat plasma, M4 had the highest peak area ratio versus internal standard (AR), constituting 44.2% of the total AUC; whereas in dog plasma, M8 was the highest with 58.3% of the total AUC (Table 3). In both rat and dog plasma, M7 was the second highest metabolite (Table 3).

Following a single *i.v.* dose of SP-8356 in rats, a total of five metabolites (M4, M5 and M7–M9) were detected in urine (Figure 3). The ARs of all the detected metabolites in the urine collected during the initial 4 h post-dosing were significantly higher than those in the urine samples collected subsequently (Figure 3). Urinary excretion of the metabolites appeared to be mostly completed during the initial 8 h after the dose, as the metabolite peaks were very small in the (8–24) h urine sample (Figure 3). In total, M8 peak was the largest, followed by M9, with the parent peak being almost insignificant, compared to the metabolite peaks in the urine samples (Figure 3). It is notable that all the urinary metabolites detected were either sulfate (M8 and M9) or glucuronide (M4, M5 and M7) conjugates.

### 2.3. Bioactivation of SP-8356 in Rat Liver S9 Fraction

Bioactivation of SP-8356 was investigated in vitro using rat liver S9 fraction fortified with various cofactors, because metabolites that could be formed by GSH conjugation of a reactive metabolite of SP-8356 were detected in vivo in rat and dog plasma (Table 2, Scheme 1).

Only ~ 20% of SP-8356 was remaining following 60 min incubation in rat liver S9 fraction without any cofactor, whereas the addition of the antioxidant ascorbic acid (AA) prevented the compound loss completely during the incubation (Table 4), suggesting that the compound loss was likely due to oxidation, such as autooxidation. NADPH as a cofactor had no significant effect, indicating the lack of involvement of CYP in the metabolism of SP-8356 in rat liver S9 fraction (Table 4). M11 was formed in the presence of GSH, which was inhibited almost completely by the addition of AA (Table 4), again suggestive of the oxidative bioactivation of SP-8356. The addition of NADPH together with GSH did not prevent the formation of M11 consistent with the lack of CYP involvement mentioned above. Cofactors for phase II metabolic reactions, UDPGA, PAPS and SAM, significantly accelerated the metabolism of SP-8356 with only 1.3% remaining following the incubation; at the same time, they inhibited M11 formation by 96% (Table 4), suggesting that the bioactivation of SP-8356 could be prevented almost completely by these competing metabolic reactions.

### 2.4. Metabolic Stability in Rat, Dog and Human Liver S9 Fractions

The metabolic stability of SP-8356 was determined in rat, dog and human liver S9 fractions supplemented with NADPH, UDPGA, PAPS, SAM and GSH.

In the absence of the cofactors, SP-8356 disappeared rapidly with *t*_1/2_ < 30 min in all three species (Figure 4, Table 5), consistent with the observation described in Section 2.3. In the presence of the cofactors, SP-8356 was metabolized more rapidly with *t*_1/2_ < 3 min in all three species (Figure 4, Table 5). The hepatic clearance (CL_H_) values predicted by the ‘well-stirred model’ using these in vitro metabolic stability data were almost identical to the hepatic blood flow of each species; therefore, the calculated hepatic extraction ratio (E_H_) values were close to unity in all species (Table 5). These results are consistent with the in vivo observations described above that SP-8356 had high CL and poor oral absorption (Figure 1, Table 1). In the presence of the cofactors, M4, M7 and M8 were detected following 60 min incubation of SP-8356 in human liver S9 fraction, with M4 constituting 52% of the total metabolite AR (data not shown); this semi-quantitative metabolite profile is similar to the in vivo results in rats (Figure 2; Table 3), suggesting that rats may be better suited to animal toxicity studies than dogs.

## 3. Discussion

The present study demonstrated that SP-8356 had high systemic CL and poor oral absorption in both rats and dogs. Consistent with these in vivo results, SP-8356 was shown to be rapidly metabolized in vitro in the animal species. Various metabolites were identified in vivo in plasma and urine.

SP-8356 was extensively metabolized in rats and dogs via various metabolic reactions, such as monooxygenation, demethylation, glucuronidation, sulfation, methylation and oxidative bioactivation, followed by GSH conjugation to a number of metabolites. Multiple metabolic pathways of SP-8356 imply a low possibility of being a victim of drug–drug interaction although it may act as a competitive inhibitor for various metabolic reactions [13]. Phase II metabolism appears to constitute a major elimination pathway of SP-8356, as among all the metabolites identified, the conjugate metabolites were most abundant in both plasma and urine. In fact, many phenol-containing compounds have been reported to be rapidly metabolized to phase II metabolites [14]. For example after oral consumption in humans, the phenolic antioxidant epicatechin was metabolized predominantly to glucuronide, sulfate and methylated conjugates [15].

A significant species difference was observed in the circulating metabolite profile of SP-8356, as in rats, glucuronide (M4) was the most abundant metabolite, while in dogs, the sulfate (M8). Species differences in the pattern of conjugation of phenolic compounds between sulfation and glucuronidation have previously been reported [16]. In mice, rats and rabbits, the cardiac stimulant xamoterol was exclusively metabolized predominantly by glucuronidation, while in dogs, appreciable amount of sulfate metabolite was formed [17]. In dog intestine, sulfation of the analgesic agent salicylamide dominated glucuronidation; while in rabbit intestine, glucuronidation was predominant [18]. Among mice, rats, dogs and humans, three representative monohydroxyflavones had the highest sulfation activity in dog liver S9 fraction [19]. It appears that in dogs, sulfation is more important for many phenolic drugs than in other species.

The molecular properties of SP-8356, including molecular weight (300.35 g/mol), clogP (~2.5), topological polar surface area (67 Å^2^), number of hydrogen bonds (2 donors and 2 acceptors) and number of rotatable bonds (3), conformed to the common rules of compounds for a high possibility of good oral bioavailability [20,21]. Moreover, the favorable theoretical solubility and permeability of SP-8356 can be predicted based on empirical formulae using the physicochemical properties [22,23]. The in vitro metabolism studies with rat and dog liver S9 fractions predicted high CL_H_ for SP-8356 that was nearly equivalent to the hepatic blood flow. The activities of metabolizing enzymes in the small intestine are in general greatly lower than in liver, based on their metabolic activities and protein contents [24]. Therefore, the poor oral absorption of SP-8356 demonstrated in this study is likely mainly due to extensive hepatic first-pass metabolism. Moreover, based on the in vitro metabolism results reported here, the pharmacokinetic properties of SP-8356 in humans are predicted to be similar to those in rats and dogs (Table 5).

GSH conjugate (M11) and its downstream metabolites (M12–M14) were detected in plasma as SP-8356 metabolites in rats and dogs, indicating that SP-8356 was bioactivated to an electrophilic reactive metabolite in the body. This was reproduced in vitro where the same GSH conjugate was formed when SP-8356 was incubated with rat liver S9 fraction in the presence of GSH. This bioactivation appears to occur via an oxidation reaction, as it was completely inhibited by the antioxidant AA. Autoxidation, as well as some other metabolic reactions in the liver, may be responsible for the bioactivation of SP-8356 [25,26]. Catechol is a well-known ‘structural alert’ that upon oxidative bioactivation, can covalently bind to endogenous nucleophiles, such as GSH and biomacromolecules (e.g., proteins and DNA) [27]. Catechol is susceptible to oxidation to form reactive OQ intermediate. OQs, as Michael acceptors, can form covalent bonds with cellular nucleophiles, including GSH, proteins and DNA, resulting in disruption of the cellular integrity and function [28]. Various catechol-containing compounds, including dihydroxycinnamic acids, are known to form GSH conjugates in oxidative conditions [29]. The catechol moiety of SP-8356 may also undergo oxidation to yield an OQ, a reactive metabolite that is subsequently conjugated with GSH to form M11; however, the OQ intermediate could not be detected, probably due to its high reactivity and instability.

Nevertheless, the potential safety issues related to SP-8356 bioactivation may not be serious, because there are competing metabolic pathways. The significance of bioactivation related to idiosyncratic toxicity is dependent not only on the daily dose but also on the existence of competing metabolic pathways [30]. The greater the contribution of competing metabolic pathways to the total CL, the lesser the fraction of the bioactivation pathway and therefore, the lesser possibility of toxicity. SP-8356 was shown to be rapidly metabolized by phase II biotransformation reactions with minimal GSH conjugate formed, suggesting that phase II reactions other than GSH conjugation, such as glucuronidation, sulfation and methylation, were responsible. The same may be said as well in vivo, because GSH conjugation-related metabolites (M11–M14) constituted only a small fraction of the total metabolite AR in the plasma. In addition, GSH conjugation itself is a detoxification pathway that can protect cells from bioactivation-related damage [31]. In fact, several catechol-containing drugs have been approved by the Food and Drug Administration (FDA), such as the selective β-adrenergic agonists isoetharine and dobutamine for treating heart failure and tolcapone for Parkinson’s disease [32].

The beneficial health effects of polyphenol-containing compounds are usually confounded by their low bioavailability and poor pharmacokinetics due to rapid metabolism to glucuronide and sulfate conjugates, which are predominant metabolites in plasma [33]. Similarly, SP-8356 exhibited a potent anti-breast cancer effect in mice, while its plasma exposure was low [10]. Several hypotheses have been proposed to interpret this discrepancy. One possibility is that conjugates may possess similar potency to the parent against the target [34,35]. For example, morphine-6-glucuronide has analgesic activity and in reality, the therapeutic effect of morphine is mainly provided by morphine-6-glucuronide [36]. Also, approximately half of the therapeutic activity of the diuretic drug triamterene was contributed by its major circulating metabolite 4-hydroxytriamterene sulfate [37]. Another explanation is that the biological activity may arise from the local deconjugation of conjugate metabolites to the bioactive parents, as in the case of flavonoids [38,39,40,41] and resveratrol [42]. Based on these previous reports, the unfavorable preclinical pharmacokinetics of SP-8356 reported here did not preclude it from further development. In fact in mice, SP-8356 glucuronide exerted similar anti-breast cancer activity to the parent [10].

In conclusion, SP-8356 exhibited high systemic CL and low oral absorption, both in rats and dogs. Rapid phase II metabolism appears to be mainly responsible for its suboptimal pharmacokinetics. Its human pharmacokinetics is predicted to be similar to those of rats and dogs. Potential safety risks related to SP-8356 bioactivation may be low, because of the competing metabolic pathways.

## 4. Materials and Methods

### 4.1. Materials

SP-8356 and (1*S*,5*R*)-4-((*E*)-4-hydroxystryryl)-6,6-dimethylbicyclo[3.1.1]hept-3-en-2-one (HSDH) were provided by Shin Poong Pharm. Co., Ltd. (Ansan, Korea). Rat (male SD rats), dog (male Beagle dogs) and human liver S9 fractions (150 donors, mixed gender) were purchased from BD Biosciences (Bedford, MA, USA). NADPH, UDPGA, PAPS, GSH, SAM, formic acid, dimethylsulfoxide (DMSO), dimethylacetamide (DMA), ethanol, Cremophor EL, Solutol HS 15 and AA were purchased from Sigma-Aldrich (St. Louis, MO, USA). Saline was purchased from JW Pharmaceutical (Seoul, Republic of Korea). HPLC grade water and acetonitrile were purchased from J.T.Baker Chemical (Radnor, PA, USA). All reagents had the highest purity available.

### 4.2. Pharmacokinetic Studies in Rats

Animal studies were approved by the Institutional Animal Care and Use Committee (IACUC) of Shin Poong Pharm. Co., Ltd. (Approval No.: SP2016-3; Approval date: Mar. 14, 2016). SP-8356 dissolved in DMSO/Cremophor EL/saline (5/10/85 vol.%) were dosed *i.v.* or *p.o.* at 20 mg/kg (*n* = 6/dose group) to male SD rats (8 weeks old; (255–275) g; Koatech Inc., Gyeonggi, Republic of Korea). Blood samples were collected at 1, 5, 10, 15, 30 min, 1, 2, 4, 6, 8 and 24 h post-dosing through femoral artery cannula into heparinized tubes containing AA (final concentration: 0.4 *w*/*v*%). The blood samples were then centrifuged at 3000 rpm at 4 °C for 10 min. The separated plasma samples were kept frozen at −80 °C, until analysis. Plasma samples (45 µL) were added with 15 µL of 50 vol.% aqueous acetonitrile and 135 µL of ice-cold acetonitrile containing HSDH (1 μg/mL) as internal standard (IS). Following a brief vortexing and sonication, the mixtures were centrifuged at 3000 rpm at 4 °C for 20 min. The supernatant (80 µL) mixed with the same volume of deionized water on an Agilent 96-well plate was subjected to analysis using Agilent 6460 Triple Quadrupole (QQQ) LC-MS/MS system equipped with dual AJS ESI ion source and Agilent 1200 series HPLC system, as follows.

Separation was performed on a Phenomenex Kinetex C18 column (2.1 mm × 50 mm, 2.6 µm) with 0.1 vol.% formic acid in water (A) and 0.1 vol.% formic acid in acetonitrile (B) as the mobile phase. The samples were analyzed in a linear gradient elution at a flow rate of 0.45 mL/min at 40 °C: starting with 5 vol.% B for 2 min, ramping to 95 vol.% B in 1 min, maintaining for 3 min and then returning to the initial condition. The injection volume was 5 µL. Ion source parameters were optimized as follows: capillary voltage, 3500 V; drying gas, 12 L/min and 345 °C; nebulizer gas, 35 psi; sheath gas, 11 L/min and 400 °C. Mass spectra were acquired in negative multiple reaction monitoring (MRM) mode at *m*/*z* (299 > 283) (F, 100 V; CE, 25 eV) and (253 > 143) (F, 100 V; CE, 30 eV) for SP-8356 and HSDH, respectively. The data were analyzed using Agilent MassHunter Quantitative Analysis software (version B.05.00). The retention times of SP-8356 and HSDH were (4.10 and 4.17) min, respectively. The linear calibration range of SP-8356 was (31.25–4000) ng/mL (r^2^ > 0.99). The precision and accuracy of the assay were evaluated using quality control samples prepared in triplicate at (30, 300 and 3000) ng/mL as described in previous report [43]. Both the accuracy and precision of this analytical method were within the acceptable ranges [43,44]. Pharmacokinetic parameters were calculated by noncompartmental analysis using PKSolver [45].

### 4.3. Pharmacokinetic Studies in Dogs

Animal studies were conducted at KNOTUS Co., Ltd. (IACUC Approval No.: KNOTUS IACUC 18-KE-363; Approval date: 24 October 2018). SP-8356 dissolved in DMSO/ethanol/Solutol HS 15/deionized water (5/3/40/52 vol.%) was dosed *i.v.* or *p.o.* at 50 mg/kg (*n* = 6/dose group) to male Beagle dogs (11 months old; (11.6–12.4) kg; OrientBio Inc., Gyeonggi, Republic of Korea). Blood samples were collected at selected time points for 24 h post-dosing through jugular vein cannula into EDTA-treated tubes containing AA (final concentration: 0.4 *w*/*v*%). The blood samples were then centrifuged at 3000 rpm at 4 °C for 20 min. The separated plasma samples were kept frozen at −80 °C, until analysis. Sample and data analysis methods were the same as those described in Section 4.2.

### 4.4. In Vivo Metabolite Identification

Plasma samples from the *i.v.* administrations described in Section 4.2 and Section 4.3 were used in this study. Plasma samples pooled at each time point were subjected to sample preparation as described in Section 4.2. The prepared samples were analyzed by Agilent 6530 Q-TOF LC-MS/MS system equipped with dual AJS ESI ion source and Agilent 1200 series HPLC system as described below.

Rat urine samples were collected from the animal studies approved by the IACUC of Korea University (Approval No.: KUIACUC-2017-124; Approval date: Jul. 29, 2017). SP-8356 dissolved in DMA/Cremophor EL/saline (5/10/85 vol.%) was dosed *i.v.* at 10 mg/kg (*n* = 5/dose group) to male SD rats (8 weeks old; (255–275) g; Koatech Inc., Gyeonggi, Republic of Korea) housed individually in metabolic cages. Urine samples were collected for ((0–4), (4–8) and (8–24)) h. Pooled urine samples were mixed with equal volume of acetonitrile containing HSDH (1 μg/mL) as IS, followed by filtration with a syringeless filter device (Mini-UniPrep™, PTFE filter media with polypropylene housing, 0.45 µm). The filtrate was analyzed by Agilent 6530 Q-TOF LC-MS/MS system equipped with dual AJS ESI ion source and Agilent 1200 series HPLC system, as follows.

Chromatographic separation was achieved on an Agilent Eclipse C18 column (2.1 mm × 100 mm, 3.5 µm) by a gradient elution with a mobile phase consisting of 0.1 vol.% formic acid in water (A) and 0.1 vol.% formic acid in acetonitrile (B) at 50 °C: starting with 5 vol.% B for 2 min, ramping to 95 vol.% B over 1 min, maintaining for 3 min. The mobile phase flow rate was kept at 0.45 mL/min throughout the run. The sample injection volume was 5 µL. Ion source parameters were set as follows: capillary voltage, 3500 V; drying gas, 12 L/min and 300 °C; nebulizer gas, 30 psi; sheath gas, 12 L/min and 300 °C. Mass spectra were acquired in negative ion auto MS/MS mode (F, 135 V; CE, 25 eV) with full scan range of *m*/*z* (100–1000) and mass scan rate of 8 spectra/s. Metabolites were identified using Agilent MassHunter Metabolite ID software (version B.04.00) (Santa Clara, CA, USA). Chromatograms and mass spectra of the parent and identified metabolites were extracted using Agilent MassHunter Qualitative Analysis software (version B.05.00) (Santa Clara, CA, USA).

### 4.5. Bioactivation Studies in Rat Liver S9 Fraction

SP-8356 was incubated with 1 mg/mL rat liver S9 fraction in 0.1 M potassium phosphate buffer (pH 7.4) in the absence and presence of GSH, AA, NADPH, UDPGA, PAPS and SAM. The final reaction mixture contained 5 µM SP-8356, 8 mM MgCl_2_, 25 µg/mL alamethicin and 1 mM cofactor in final volume of 160 µL. The reaction was initiated by adding a 10 mM cofactor solution prepared in 0.1 M potassium phosphate buffer (pH 7.4), following a 5 min preincubation of the incubation mixture at 37 °C. The final study groups were as follows: no cofactor (control), AA, NADPH, GSH, GSH + AA, GSH + NADPH and GSH + UDPGA + PAPS + SAM. The reaction was quenched at (0 and 60) min by adding 160 µL of ice-cold acetonitrile containing HSDH (1 µg/mL). Following a brief vortexing and sonication, the mixtures were centrifuged at 3,000 rpm at 4 °C for 20 min. The supernatant mixed with the same volume of deionized water on an Agilent 96-well plate was analyzed by Agilent 6530 Q-TOF LC-MS/MS system equipped with dual AJS ESI ion source and Agilent 1200 series HPLC system, as follows.

Chromatographic separation was performed on a Phenomenex Kinetex C18 column (50 mm × 2.1 mm, 2.6 µm) (USA) with 0.1 vol.% formic acid in deionized water (A) and 0.1 vol.% formic acid in acetonitrile (B) running at 0.45 mL/min in the following gradient elution: 2 vol.% B for the initial 2 min, linear gradient to 95 vol.% B for 1 min and maintaining 95 vol.% B for 5 min. The injection volume was 5 μL and the column temperature was set at 50 °C. The ion source and mass analyzer parameters of the mass spectrometer were the same as in Section 4.4. LC-MS chromatograms and mass spectra were extracted using Agilent MassHunter Qualitative Analysis software (version B.05.00) (Santa Clara, CA, USA).

### 4.6. Metabolic Stability in Liver S9 Fractions

The metabolic stability of SP-8356 was investigated using rat, dog and human liver S9 fractions fortified with cofactors for major drug metabolism. The final reaction mixture contained 1 mg/mL rat, dog or human liver S9 fraction, 1 mM cofactors (NADPH, UDPGA, PAPS, SAM and GSH), 2 mM MgCl_2_, 25 µg/mL alamethicin and 1 µM SP-8356 in final volume of 200 µL. The metabolic reactions were initiated by adding 10 mM cofactor solution to reaction mixtures preincubated for 5 min at 37 °C. The reactions were terminated at (0, 5, 15, 30 and 60) min by the addition of 200 µL ice-cold acetonitrile containing HSDH (1 µg/mL). The subsequent sample preparation and analytical methods were the same as those described in Section 4.5.

The degradation *t*_1/2_ of SP-8356 was calculated using the following Equation:(1)$$t_{1/2}=\frac{0.693}{k},$$
where, *k* was the first-order rate constant calculated by one phase decay nonlinear regression using GraphPad Prism™ 7 (GraphPad Software, La Jolla, CA, USA). CL_int_ was calculated using the following Equation:(2)$$CL_{int}=\frac{k}{f_{u,S9}}\cdot \;\frac{mL\;incubation}{mg\;liver\;S9}\cdot \;\frac{mg\;S9}{g\;liver}\cdot \frac{g\;liver}{kg\;body\;wt}.$$

The scaling factors were (136, 120.7 and 120.7) mg *S*9 protein/g liver and (40, 32 and 26) g liver/kg body weight for rats, dogs and humans, respectively [46,47]. For simplicity, the fraction unbound in *S*9 incubation (*f*_u, S9_) was assumed to be one. CL_H_ was predicted using the hepatic well-stirred model, as follows:(3)$$CL_{H}=\frac{Q_{H}\cdot f_{u}\cdot CL_{int}}{Q_{H}+f_{u}\cdot CL_{int}},$$
where, *Q*_H_ was the hepatic blood flow of (55, 30 and 20) mL/min/kg for rats, dogs and humans, respectively. The fraction unbound (*f*_u_) was assumed to be one. *E*_H_ was calculated using the following Equation:(4)$$E_{H}=\frac{CL_{H}}{Q_{H}}.$$

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
