# Peer review of "Metabolism and Pharmacokinetics of SP-8356, a Novel (1S)-(−)-Verbenone Derivative, in Rats and Dogs and Its Implications in Humans"

_molecules, 2020, doi:10.3390/molecules25081775_

Round 1
Reviewer 1 Report
This study by Zhou et al. reported the metabolism and pharmacokinetic study of SP-8356, a drug candidate undergoing preclinical development for the treatment of ischemic stroke and atherosclerosis. By using mass spectrometry approach, they profiled the pharmacokinetic property of SP-8356 and also identified quite a few metabolites (fourteen in total) in rats and dogs. These data provide useful information for further moving this drug candidate to the next stage. Overall the presented data is of high quality and interpretations drawn are sound. I recommend publication of this manuscript in its current form without reservations.
Author Response
We appreciate the comments and suggestions of the reviewer.
No responses are provided as the reviewer recommended publication of this manuscript in its current form.
Reviewer 2 Report
Title: Metabolism and Pharmacokinetics of SP-8356, a Novel (1S)-(-)-Verbenone Derivative, in Rats and Dogs, and its Implications in Humans
the paper in interesting and only minor concerns need to be addressed, as follow:
Page1, Line 18: A bracket is lacking in the exact name of the molecule. Please, check and correct.
Page 1, Line 23: “half-life” instead of “half-live”. Please, correct.
Furthermore, consider to indicate the common abbreviation “t1/2”.
Page 1, line 71: How did you characterized the metabolites? Which analytical technique did you apply?
Page 1, line 72: Why did you chose the S9 fractions and what are the cofactors used?
Page 2, Line 82 and 87: Since you stated that a single dose was administered to the animals, it is more correct to use the singular “dose” or the term “administration”.
Page 2, Line 83: Please, put the number out of the brackets
Page 2, Line 91: It is not clear if you assessed one metabolite or a set of molecules. What is the metabolite assessed in your study? Are more than one?
Page 3, Line 107: Please, correct the word metabolites.
Page 8, Line 228: Agreeing with you in considering the multiple metabolic pathways of a drug an advantage, it has to be considered the pharmacokinetic interaction with a larger number of drugs. This aspect should be evaluated and discussed.
Page 8, line 255: Please, provide references.
Page 10, line 318: What is the time schedule applied for the collection of samples? what is the rationale at its base?
Page 10, line 337-340: Was the method validated before the analysis? What was the validation protocol applied? Please, provide the validation parameters considered and the references taken into account for acceptance criteria.
Page 11, line 359: What is the time schedule followed for collecting samples?
Page 11, line 372: Please, specify in brackets the Country of provenience of the company Agilent
Page 11, line 377: How did you prepare the cofactors mixture? What was the concentration of each cofactor in the final mixture? What solvent did you use for preparing this mixture?
Page 11, line 387: Please, specify in brackets the Country of provenience of the company Phenomenex
